# Building a Cell House from Cellulose: The Case of the Soil Acidobacterium *Acidisarcina polymorpha* SBC82^T^

**DOI:** 10.3390/microorganisms10112253

**Published:** 2022-11-14

**Authors:** Svetlana E. Belova, Daniil G. Naumoff, Natalia E. Suzina, Vladislav V. Kovalenko, Nataliya G. Loiko, Vladimir V. Sorokin, Svetlana N. Dedysh

**Affiliations:** 1Winogradsky Institute of Microbiology, Research Center of Biotechnology of the Russian Academy of Sciences, 119071 Moscow, Russia; 2Skryabin Institute of Biochemistry and Physiology of Microorganisms, Pushchino Scientific Center for Biological Research of the Russian Academy of Sciences, Moscow Region, 142290 Pushchino, Russia; 3N.N. Semenov Federal Research Center for Chemical Physics, Russian Academy of Sciences, 119991 Moscow, Russia

**Keywords:** cellulose biosynthesis, cellulose synthase, GT2 family, *bcsAB* gene, *Acidisarcina polymorpha*, *Acidobacteriota*, protein phylogenetic tree, lateral gene transfer, non-orthologous gene displacement, X-ray scattering

## Abstract

*Acidisarcina polymorpha* SBC82^T^ is a recently described representative of the phylum *Acidobacteriota* from lichen-covered tundra soil. Cells of this bacterium occur within unusual saccular chambers, with the chamber envelope formed by tightly packed fibrils. These extracellular structures were most pronounced in old cultures of strain SBC82^T^ and were organized in cluster-like aggregates. The latter were efficiently destroyed by incubating cell suspensions with cellulase, thus suggesting that they were composed of cellulose. The diffraction pattern obtained for 45-day-old cultures of strain SBC82^T^ by using small angle X-ray scattering was similar to those reported earlier for mature wood samples. The genome analysis revealed the presence of a cellulose biosynthesis locus *bcs*. Cellulose synthase key subunits A and B were encoded by the *bcsAB* gene whose close homologs are found in genomes of many members of the order *Acidobacteriales*. More distant homologs of the acidobacterial *bcsAB* occurred in representatives of the *Proteobacteria*. A unique feature of *bcs* locus in strain SBC82^T^ was the non-orthologous displacement of the *bcsZ* gene, which encodes the GH8 family glycosidase with a GH5 family gene. Presumably, these cellulose-made extracellular structures produced by *A. polymorpha* have a protective function and ensure the survival of this acidobacterium in habitats with harsh environmental conditions.

## 1. Introduction

The *Acidobacteriota* (former name *Acidobacteria*) is one of the phylogenetically diverse and cosmopolitan phyla of the domain Bacteria [1,2,3,4,5]. Known diversity within this phylum includes 26 major 16S rRNA gene sequence clades or subdivisions (SDs) [2], which were assigned to 15 class-level groups, five of which contain taxonomically described representatives [6]. Members of the phylum *Acidobacteriota* colonize a large variety of terrestrial habitats with highly diverse environmental conditions, such as pristine and agricultural soils, wetlands of different trophic status, caves, unfixed aeolian sands, freshwater and subsurface sediments, hot springs, and many other ecosystems [2,5,7,8,9,10,11,12,13]. Soils of all climatic zones, from sandy soils of arctic tundra to organic-rich soils of tropical forests, are one of the major habitats of these bacteria. As evidenced by the results of cultivation-independent microbial diversity surveys, *Acidobacteriota*-affiliated sequences comprise between 5 and 50% of 16S rRNA gene reads commonly retrieved from various soils [7,8,10,13,14,15,16,17]. Notably, soils colonized by the acidobacteria differ dramatically regarding their physicochemical characteristics, such as temperature, pH, moisture, contents of organic carbon, phosphorous, and available nitrogen, availability of plant-derived organic matter, mineralogical composition, and other ecologically important parameters. As shown in a recent study that employed a culture-independent niche modeling approach to elucidate ecological adaptations and their evolution for over 4000 operational taxonomic units (OTUs) of *Acidobacteriota* across 150 different, comprehensively characterized grassland soils, the high diversity of soil acidobacterial communities is largely sustained by differential habitat adaptation even at the level of closely related species [13].

The highest relative abundances of *Acidobacteriota* are commonly observed in acidic soils and peatlands, which are dominated by representatives of SD1 and 3 [8,9,11,18,19,20]. Thus, our previous study of microbial communities in lichen-covered tundra soil in northwest Siberia, Russia, revealed acidobacteria as one of the major bacterial groups (22–24% of all classified 16S rRNA gene sequences) in this harsh environment characterized by low pH and low temperatures [21]. Between 59 and 71% of total *Acidobacteriota*-like reads retrieved in that study belonged to members of SD1. A representative isolate of these bacteria, strain SBC82^T^, was obtained in a pure culture and assigned to a novel genus and species *Acidisarcina polymorpha* [21]. Taxonomic analysis revealed that the sequences affiliated with the strain SBC82^T^ at the 97% similarity level comprised 5–17% of all *Acidobacteria*-like reads.

Colonies of *Acidisarcina polymorpha* SBC82^T^ had gummy consistency and were composed of polymorphic cells that were arranged in sarcina- or cluster-like aggregates [21]. Cells of this acidobacterium occurred within unusual saccular chambers, with the chamber envelope formed by tightly packed fibrils. The structure and chemical composition of these unusual extracellular structures were not examined. Strain SBC82^T^ utilized various sugars and polysaccharides and grew in a wide range of pH (4.0–7.7) and temperatures (5–36 °C). The genome of strain SBC82^T^ consisted of a 7.11-Mb chromosome and four megaplasmids, and encoded a wide repertoire of enzymes involved in the degradation of chitin, cellulose, and xylan. The ability to utilize amorphous chitin as a source of carbon and nitrogen was proved experimentally, thus making *Acidisarcina polymorpha* SBC82^T^ the first acidobacterium with experimentally confirmed chitinolytic potential.

The initial genome analysis also suggested the occurrence of genetic determinants for cellulose biosynthesis in strain SBC82^T^, but no detailed analysis has been made in the original study [21]. The evidence for cellulose biosynthesis was noted in several earlier studies that analyzed genomes of SD1 acidobacteria [22,23], but none of these studies included experimental verification of this genome-encoded capability.

Bacterial cellulose is an exopolysaccharide produced by different bacterial species, including both Gram-negative and Gram-positive bacteria [24,25]. In natural habitats, this exopolysaccharide is produced to form protective envelopes around the cells [26]. The celluloses produced by different bacteria possess different structure, properties, and biological roles. Among the latter, the roles of cellulose in attachment of bacterial cells to plants [27], flocculation in wastewater [28,29], high water uptake/holding capacity [30] are well recognized.

The uridine diphosphate-forming form of cellulose synthase (EC 2.4.1.12) is the main enzyme that produces cellulose. It has been found in Bacteria and Eukaryota (alga, plants, Urochordates), but all eukaryotic genes most probably have chloroplast origin [31,32,33]. Cellulose synthase represents a vast group of enzymes known as glycosyltransferases. Glycosyltransferases (GT) and glycoside hydrolases (GH) are the main enzymes for the synthesis and utilization of carbohydrates, respectively. Based on amino acid sequence homology, they are grouped into families in the CAZy database [34]. To date, 114 families of glycosyltransferases and 165 families of glycoside hydrolases are known. Five glycoside hydrolase families (GH5, GH13, GH16, GH30, and GH43) are divided into subfamilies at a lower hierarchical level. In particular, the GH5 family consists of 56 subfamilies (GH5_1–GH5_56).

Cellulose synthesis in bacteria is controlled by the *bcs* locus, which has been studied in detail in a number of *Proteobacteria*. The key component of this locus is the *bcsA* gene encoding the inner-membrane catalytic subunit A of cellulose synthase (EC 2.4.1.12), which contains the cytosolic domain of the family 2 of glycosyltransferases (GT2). It is generally accepted that this gene is the first one in the *bcsABCD* operon [35]. The next *bcsB* gene encodes the periplasmic B subunit with a carbohydrate-binding domain. It is indispensable for cellulose polymerization, thus making BcsB a co-catalytic subunit. A number of *Proteobacteria* synthesize the two-domain BcsAB protein. The tetratricopeptide repeat-rich BcsC protein is located in the outer membrane and acts as a porin. The BcsD protein is facultative and contributes to the regular packing of glucan chains in species secreting crystalline cellulose. The same locus hosts the *bcsZ* gene encoding periplasmic cellulase, which belongs to the GH8 family of glycoside hydrolases. Additional Bcs components, such as BcsH, BcsO, BcsP, BcsQ, BcsR, or BcsS can also be encoded. The neighboring *bcsEFG* operon is also involved in cellulose synthesis. The structure and composition of the *bcs* locus vary greatly in different *Proteobacteria* [35,36,37,38,39]. Cellulose synthesis loci have also been found in *Acidobacteriota* [23], *Actinobacteria* [40], *Cyanobacteria* [36], and *Firmicutes* [41]. *Clostridioides difficile* cellulose synthesis locus has *ccsZ* gene encoding the GH5_25 subfamily endoglucanase (GenPept, AJP12293.1).

In this study, we followed up the formation of the unusual extracellular saccular chambers produced by *Acidisarcina polymorpha* SBC82^T^ to elucidate their nature and the potential functional role. We also performed a detailed analysis of the cellulose synthase locus in the genome of strain SBC82^T^ and compared it to that in other described acidobacteria.

## 2. Materials and Methods

### 2.1. Strain and Culture Conditions

*Acidisarcina polymorpha* SBC82^T^ (=KCTC 82304^T^ = VKM B-3225^T^) was used in this study. Strain SBC82^T^ was maintained on the solid medium MA containing (per liter distilled water): 0.5 g fructose, 0.1 g yeast extract, 0.2 g KNO_3_, 0.04 g MgSO_4_⋅7H_2_O, 0.04 g KH_2_PO_4_, 0.02 g CaCl_2_⋅2H_2_O, 9 g phytagel, 0.03 g alginic acid, pH 4.6–5.0. For the electron microscopy, X-ray microanalysis, and the analysis using small angle X-ray scattering, strain SBC82^T^ was grown in the liquid medium MA with shaking at 180 rpm under room temperature.

### 2.2. Examination of Cell Morphology

Morphological observations of 10-, 20- and 40-day-old cells of strain SBC82^T^ were made with a Zeiss Axioplan 2 microscope and Axiovision 4.2 software (Zeiss, Germany).

For preparation of ultrathin sections, 14-day-old and 30-day-old cells of strain SBC82^T^ were collected from plates and pre-fixed with 1.5% (*w*/*v*) glutaraldehyde in 0.05 M cacodylate buffer (pH 6.5) for 1 h at 4 °C and then fixed with 1% (*w*/*v*) OsO_4_ in the same buffer for 4 h at 20 °C. After dehydration in an ethanol series, the samples were embedded into Epon 812 epoxy resin. Sections were obtained using the Reichert-Jung Ultracut ultramicrotome (Austria). Sections were mounted on the support grids, contrasted for 30 min in a 3% solution of uranyl acetate in 70% alcohol, additionally contrasted by lead citrate at 20 °C for 4–5 min, and examined in the JEM-1400 transmission electron microscope (JEOL, Japan) at accelerating voltage of 80 kV at the UNIQEM Collection Core Facility, Research Center of Biotechnology of the Russian Academy of Science.

### 2.3. X-ray Microanalysis

X-ray microanalysis of unstained ultrathin cell sections was performed using a JEM-1400 microscope (JEOL, Akishima, Japan) equipped with an X-ray microanalyzer INCA Energy TEM 350 EDS system (Oxford Instruments) at the UNIQEM Collection Core Facility, Research Center of Biotechnology of the Russian Academy of Science.

### 2.4. Small Angle X-ray Scattering (SAXS) and Data Processing

In the study, 14- and 45-day-old cultures of strain SBC82^T^ were used for small-angle X-ray scattering detection. Similar culture aliquots were centrifuged at 10,000× *g* for 5 min. A thin layer of cellular biomass was placed in the special mesh sample holder, which was installed opposite the X-ray beam.

SAXS measurements were performed at synchrotron stations ID23-1 and ID23-2 at the European Synchrotron Radiation Facility—ESRF (Grenoble, France), with the following parameters: Wavelength—2 Å, beam spot—4 microns, flux—5 × 10^10^ photons/second, exposition time—5 s per shot. Scattered radiation was recorded by Pilatus 6M and Pilatus 3X 2M detectors, placed at a 982.8 mm distance behind the sample holder. All measurements were held at 100 K temperature. The scheme of the SAXS experiment is depicted in Figure 1A.

For a detailed analysis of the angle distribution of individual fibrils respectively to each other f(μ), values of radial averaged intensity profile H(χ) were calculated by using Equation (1). The relationship between H(χ) and f(μ) is shown in (2) and Figure 1B:(1)$$H\left(\chi \right)=\int_{q_{min}}^{q_{max}}I\left(q_{r},\chi \right)dq_{r}$$
(2)$$H\left(\chi \right)=4\int_{\chi }^{\pi /2}\frac{\;f\left(\mu \right)sin\left(\mu \right)}{\sqrt{cos^{2}\chi -cos^{2}\mu }}d\mu$$

### 2.5. Enzymatic Treatment of Cells of Strain SBC82^T^ with Cellulase

Cellulase (other names: endo-1,4-β-glucanase, 1,4-(1,3:1,4)-β-D-glucan-4-glucanohydrolase) (Serva Feinbiochemica, Heidelberg, Germany) was used to treat the cellular aggregates of strain SBC82^T^. For this, 6 mg cellulase (1.55 U/mg) was dissolved in 300 μL of 20 mM acetic buffer (pH 4.5). The resulting cellulase solution (0.03 U/µL) was used to treat the cells collected from 45-day-old culture of strain SBC82^T^ at 40 °C for 1–2 h. The results of this treatment were verified by using phase-contrast microscopy.

### 2.6. Phylogenetic Analysis

The full-length amino acid sequence of the BcsAB protein from *Acidisarcina polymorpha* SBC82^T^ (ACPOL_3662, AXC12945.1) was used to search for close homologs using the blastp algorithm on the NCBI website (http://www.ncbi.nlm.nih.gov/ (accessed on 9 April 2022). Proteins of acidobacteria *Candidatus* Sulfotelmatomonas gaucii (GenPept, SPE19250.1) and *Granulicella aggregans* M8UP14 (MBB5059869.1) were excluded from further analysis, as they are incomplete and, presumably, correspond to pseudogenes. For the construction of a multiple alignment and further phylogenetic analysis, 74 nearest (almost) full-length homologs of ACPOL_3662 were selected. Multiple sequence alignment was prepared manually using the program BioEdit (https://bioedit.software.informer.com/7.2 (accessed on 9 April 2022) on the basis of blastp pairwise alignments.

After removing the most variable regions, the multiple sequence alignment was used to implement phylogenetic inference programs, using the maximum parsimony method. Programs SEQBOOT, PROTPARS, and CONSENSE from the PHYLIP package (http://evolution.gs.washington.edu/phylip.html (accessed on 9 April 2022) were successively used to derive confidence limits estimated by 1000 bootstrap replicates. The program TreeView Win32 (https://colab.research.google.com/drive/11lKiuyH6X2o2XifQKxiH5JjLAskO_3dX (accessed on 9 April 2022) was used for drawing the trees, while the most divergent group among the analyzed proteins (which corresponded to proteobacteria) was chosen as the outer one. The list of experimentally characterized representatives of the GT2 glycosyltransferase family was taken from the CAZy database [34].

## 3. Results

### 3.1. Cell Morphology

Examination of cells of strain SBC82^T^ at different growth stages by phase-contrast microscopy revealed highly variable cell morphology that varied from single polymorphic cells to large aggregates composed of sarcina-like cell clusters (Figure 2). At the early growth stages, the cells were odd in shape, elongated, and slightly convoluted (Figure 2a). Single-cell diameter on the cell poles could be very different at this stage of growth. Single cells and chains of different lengths consisting of curved deformed cells could also be observed occasionally (Figure 2a). Over time, the cells took the form of short rods that were arranged in sarcina-like tetrads (Figure 2b). Old cultures of strain SBC82^T^ contained large and tightly packed cell aggregates (Figure 2c).

### 3.2. Cell Ultrastructure

Electron microscopy of ultrathin cell sections of strain SBC82^T^ revealed a cell wall structure typical of Gram-negative bacteria. The cell wall consisted of a cytoplasmic membrane, a thin electron-dense peptidoglycan layer, and an outer membrane (Figure 3). Many small heterogeneous electron-dense inclusions that looked similar to polyphosphate granules were observed in the cytoplasm.

The unique ultrastructural feature of this acidobacterium, however, was the presence of extracellular structures appearing as saccular chambers, with the cells occurring and dividing inside these structures (Figure 3). The chamber envelope was composed of tightly packed fibrils layered in parallel to each other (see insert in Figure 3b). These characteristic structures were observed in cells from both young (Figure 3a) and old cultures of strain SBC82^T^ (Figure 3b), but the chamber envelope was significantly thicker in cells from old cultures. The cells occurring inside the chambers were separated from the chamber envelope by the extensive electron-transparent matrix. The latter contained a fibrillar matter of low density as well as numerous vesicles, which were released from the outer cell membrane in the chamber matrix and transferred to the inner surface of the chamber envelope. Notably, the chamber division by means of the fibrillar septum formation was also observed (Figure 3b).

X-ray microanalysis of the ultrathin cell sections revealed the presence of silicon (Si) in the chamber wall (Figure 3c).

### 3.3. X-ray Diffraction Patterns from Cells of Strain SBC82^T^

SAXS measurements have an advantage over electron microscopy due to the ability to analyze samples in the native hydrated state, which minimizes the risk of artifacts and allows quick data acquisition. This method has been used to detect the orientation distribution of cellulose microfibrils in natural objects [42]. In this work, the SAXS method was applied to analyze bacterial cell conglomerates in a 45-day-old culture of strain SBC82^T^ (Figure 4(Ia)). A 14-day-old culture consisting mainly of single cells was used as a control (Figure 4(IIa)). The diffraction pattern of the detector-recorded two-dimensional distribution of scattering intensity from a 45-day-old culture of strain SBC82^T^ was represented by symmetrical radial streaks (Figure 4(Ib)). This indicated the presence of a rod-like scattering object resembling ordered fibrils in the sample. In contrast, the scattering intensity recorded for a 14-day-old culture had circular symmetry, indicating the absence of ordered fibrils which, however, did not exclude the presence of a small amount of stochastically oriented fibrils (Figure 4(IIb)).

Further processing of the obtained diffraction data was performed to elucidate the structure of the chamber envelope produced by strain SBC82^T^. Based on the two-dimensional distributions of scattering intensity of 45- and 14-day-old cell samples, one-dimensional curves of averaged intensity depending on the distance from the detector center *I*(*q_r_*) were plotted (Figure 4(Ic,IIc)). For the 45-day-old culture, an averaging was performed only over the detector region where symmetrical streaks were recorded. The obtained curve was approximated (in the region of *q* values corresponding to intraparticle interference) by a power law decay ~*q*^−2.03^ and contained no modulations and peaks (Figure 4(Ic)). In the SAXS experiment, this decay of the scattering intensity indicates the presence of lamellar (layered) structures in the solution [43]. In our case, this indicated the layered structure of the acidobacterial chamber envelope, with each layer consisting of densely packed fibrils. Averaging of the scattering intensity for the 14-day-old culture was performed over the entire area (360-degree angle), since the two-dimensional scattering pattern possessed a circular symmetry. The one-dimensional curve of the averaged intensity dependence on the distance from the detector center clearly differed from that of 45-day-old culture (Figure 4(IIc)). The curve could not be approximated by a power decay function but had a very broad peak centered at the *q* = 0.85 nm^−1^ value. This suggested the presence of some weakly periodic structure in the examined material with a characteristic period of 7.5 nm.

To characterize the arrangement of fibrous structures in a 45-day-old culture, the obtained diffraction data were processed in a different way via averaging of two-dimensional distributions of scattering intensity by each detector pixel corresponding to the same angle *χ* (the angle is indicated in Figure 1) (*H*(*χ*)). The curve displayed in Figure 4(Id) shows the numerical expression of the sum of scattering intensity recorded by each detector pixel located along the angle *χ*. Determined *H*(*χ*) values give the possibility to calculate the positional arrangements of individual microfibrils in the chamber envelope of strain SBC82^T^.

The *H*(*χ*) plot (Figure 4(Id)) shows the intensity peak corresponding to the position of the symmetric streak in the two-dimensional diffraction pattern from a 45-day-old culture. Notably, the central part of the streak was much brighter than its surroundings. This suggests that the microfibrils in the chamber envelope deviate from each other by a small angle. The *H*(*χ*) values were used to calculate the distribution of the angle μ between two individual microfibrils (see Formula (2) in the Section 2.4.). The corresponding probability distribution curve *f*(*μ*) is shown in Figure 4(Ie). According to these calculations, the microfibrils from the chamber envelope were arranged nearly parallel to each other, because the root-mean-square deviation of μ was only 3.3°.

Analysis of the averaged intensity *H*(*χ*) curve for a 14-day-old culture, which had no peaks (Figure 4(IId)), confirmed the conclusion about the absence of fibrous structures or their stochastic arrangement relative to each other in young cells.

Thus, the obtained SAXS data are in good agreement with the observations made by using electron microscopy, which showed the occurrence of densely packed fibrous structures in the cell chambers of old cultures of strain SBC82^T^ (Figure 3). Although these fibrous structures were also present in minor amounts in cells from young cultures, they were not yet structurally organized and densely packed.

### 3.4. Enzymatic Treatment of Cell Conglomerates with Cellulase

The treatment of 45-day-old culture of strain SBC82^T^ composed of tightly packed cell conglomerates with cellulase resulted in an apparent degradation of cell clusters and related extracellular structures so that the obtained suspension was composed of single cells and small incoherent cell aggregates (Figure 5). This result provided additional evidence for the presence of cellulose in the extracellular saccular structures of strain SBC82^T^.

### 3.5. Structure of the Cellulose Synthesis Locus

According to Table S1 from the supplement to Belova et al. (2018) [21], this locus contains five genes in *Acidisarcina polymorpha* SBC82^T^: ACPOL_3660–ACPOL_3664. The ACPOL_3660 encodes GH5 family cellulase, ACPOL_3661 and ACPOL_3662 encode BcsQ and BcsAB proteins, respectively, and ACPOL_3663 and ACPOL_3664 encode two fragments of the BcsC protein (Figure 6).

Possible functionality of the BcsC protein is unclear: the corresponding genome region is considered a pseudogene by Belova et al. [21] and GenBank (CP030840.1). The structure of the *bcs* locus of *A. polymorpha* is similar to that described previously for several other representatives of acidobacteria [23], except that instead of the GH5 family glycoside hydrolase gene, other acidobacteria posses the BscZ endoglucanase (GH8 family) gene located between the *bscAB* and *bscC* genes (Figure 6).

Our screening of the NCBI database using the ACPOL_3662 protein as a query revealed that its close homologs are present in many members of the order *Acidobacteriales*, with some of them having two paralogs. Analysis of the genome context showed that the *bcsAB* gene is usually located in acidobacteria as part of the cellulose synthesis locus (Figure 6), but if there are two paralogs, then this is true for only one of them. The figure shows the structure of this locus for three such organisms, each having two *bcsAB* paralogs. The second paralogous gene is located in *Acidobacteria* bacterium AB60 (DYQ86_07050), *Granulicella rosea* DSM 18704 (SAMN05421770_10970), and *Terriglobus roseus* DSM 18391 (Terro_0419) in a variable genome context. In the case of *T. roseus*, the neighboring gene (Terro_0418) turned out to be the gene of yet another glycosyltransferase (GT4 family). Our analysis confirmed the data of Rawat et al. [23] about the high conservation of the structure of the cellulose synthesis locus among acidobacteria. The statement of Rawat et al. (2012) about the absence of the *bcsC* gene in *Acidobacterium capsulatum* ATCC 51196 appears to be incorrect since it was annotated as a pseudogene (ACP_0077) in the corresponding genome sequence (GenBank, CP001472.1).

Among the host organisms of more distant homologs of the ACPOL_3662 protein, it was possible to detect representatives of a number of proteobacteria that possess the two-domain BcsAB protein. In the case of *Escherichia coli* GE3 (GenBank, CP012376.1), the structure of the cellulose synthesis locus characteristic of acidobacteria was observed (AKN41_3846–AKN41_3849), while in *Shewanella avicenniae* FJAT-51800 (CP071503.1) there was a slightly different order and set of genes (Figure 6). In both cases, the presence of the gene encoding BcsZ endoglucanase from the GH8 family was unchanged. This indicates that it was the original component of this locus in acidobacteria but was lost in the lineage leading to *A. polymorpha* SBC82^T^. Its ACPOL_3660 protein turned out to belong to the GH5_1 subfamily, which is not characteristic of most acidobacteria: Among genome-wide sequences, its fairly close homolog (56% amino acid sequence identity) was found in *Edaphobacter aggregans* EB153 (EDE15_2663), but its gene located in the genome far from the cellulose synthesis locus (EDE15_1043–EDE15_1046).

### 3.6. Phylogenetic Analysis of Cellulose Synthase BcsAB

Multiple alignment of bacterial cellulose synthase amino acid sequences showed that they consist of two conservative regions (BcsA and BcsB), which are connected by a linker of variable size and composition. These two regions were used to reconstruct BcsAB phylogenetic tree (Figure 7). All analyzed acidobacterial proteins compose a very stable monophyletic group (95.7% bootstrap support), whose most divergent representative is ACPOL_3662 from *A. polymorpha* SBC82^T^. Paralogous BcsAB proteins encoded in eight acidobacteria genomes belong to two distinct clusters (99.8% and 99.2%) on the tree. One of them includes proteins encoded by genes located in cellulose synthase loci, but the other one contains only their outcast paralogs.

## 4. Discussion

The first evidence for exopolysaccharide biosynthesis by *Acidobacteriota* was obtained by Eichorst and co-authors [44], who isolated several representatives of these bacteria from soils and described them as representing a novel genus and species, *Terriglobus roseus*. These soil isolates produced an extracellular matrix of yet unknown chemical composition, which caused cells to stick together and form visible clumps in liquid culture. As was suggested in that study, the formation of such extracellular material in soils may serve as a form of protection from predation or as a web to trap water or nutrients and may be involved in the formation of soil aggregates [44]. Later, extensive production of an amorphous extracellular polysaccharide-like substance was described for representatives of another acidobacterial genus, *Granulicella*, which was isolated from boreal peatlands and Arctic tundra soils [16,45]. Similar observations regarding the exopolysaccharide biosynthesis were reported for many other members of the order *Acidobacteriales*, which corresponds to the formerly recognized subdivision 1 (SD 1) of the phylum *Acidobacteriota* [6]. In addition to *Granulicella* and *Terriglobus*, the list of exopolysaccharide-producing members of the *Acidobacteriales* includes the genera *Acidobacterium*, *Acidipila*, *Acidisoma*, *Bryocella*, *Edaphobacter*, and *Terracidiphilus*. The only study that characterized exopolysaccharides produced by acidobacteria was performed for two strains of the genus *Granulicella* [46]. However, only a fraction of water-soluble polysaccharides was examined in that work.

The first genome analysis of acidobacteria was performed for *Acidobacterium capsulatum* DSM 11244^T^, *Candidatus* Koribacter versatilis Ellin345, and *Candidatus* Solibacter usitatus Ellin6076 [22]. Of these, only *Acidobacterium capsulatum* DSM 11244^T^ belongs to the order *Acidobacteriales*. The *A. capsulatum* genome contained an operon with all the genes identified as necessary for cellulose synthesis and, at that time, was one of the very few non-proteobacterial genomes reported to contain this operon. The latter appeared to be incomplete in two other examined acidobacteria since the cellulose synthesis gene was not identified. Later, the genome analysis of three Arctic tundra soil acidobacteria of the genera *Granulicella* and *Terriglobus* also revealed their ability for biosynthesis of diverse structural polysaccharides, including cellulose [23]. The ability to synthesize structural carbohydrates involved in cell envelope biogenesis was suggested to be of special importance for maintaining cell integrity in tundra soil habitats that are characterized by low temperatures and multiple freeze-thaw cycles [23]. As we see from Figure 7, the ability to produce cellulose is encoded in the genomes of many representatives of the order *Acidobacteriales*, including both strains from tundra habitats and other soils. The presence of cellulose synthesis genes suggests potential traits for desiccation resistance, biofilm formation, and/or contribution to soil structure [22].

The experimental evidence for cellulose biosynthesis by acidobacteria, however, was lacking till now. In our study, the SAXS method using the synchrotron radiation source ESRF in Grenoble was applied for the first time to analyze the structure of the capsular envelope produced by the soil acidobacterium. Previously, this method was used to study the structure of wood from various plant species [42], cellulose-containing cell walls of *Arabidopsis thaliana* and the algae *Chara corallina*, and cells of *Dictyostelium discoideum*, which contain actin protein fibrils [47]. Using the SAXS method allows obtaining averaged data for an extremely small object determined by the size of the X-ray beam and, in the presence of a fibrous structure, to determine the orientation of microfibrils. Application of this method to 45-day-old cellular conglomerates produced by strain SBC82^T^ revealed the occurrence of a layered structure consisting of densely packed fibrils, presumably of cellulose nature. Calculations have shown that the packing of fibrils in the cell envelope is very dense, even denser than that in the wood of two softwood species (Norwegian spruce and Scots pine) and two hardwood species (pedunculated oak and copper beech) [42]. In contrast, the results obtained for 14-day-old culture demonstrated a lack of an orderly stacked fibril structure which, however, does not rule out the presence of fibrils in their stochastic orientation. A similar situation was described for algal cells [48] in which dense packing of cellulose fibrils in the cell wall was evident in six-day-old individuals and was not evident in young one-day-old cells. Although the streak pattern obtained in our study was clearly different from that obtained for actin-containing cells of *Dictyostelium discoideum* [47], these results served only as indirect evidence for cellulose presence in chamber envelopes of strain SBC82^T^. The treatment of 45-day-old culture of strain SBC82^T^ with cellulase and the observed degradation of cell clusters provided additional direct evidence for the presence of cellulose in the extracellular saccular structures of this soil acidobacterium.

Notably, the catalytic subunit BcsAB of cellulose synthase in *Acidisarcina polymorpha* SBC82^T^ occupies the most divergent position in the corresponding phylogenetic tree (Figure 7). The chamber-like structures described in this study have not been observed in other members of the *Acidobacteriales* as well as in other cultivated bacteria. The only study that described somewhat similar extracellular structures was focused on radiation-resistant pseudomonads [49]. In that case, the chamber-like envelopes played a protective role by integrating cells and providing vegetative growth in conditions of continuous intensive UV irradiation. The material used to construct the chamber-like structures in these pseudomonads, however, has not been analyzed.

The analysis of BcsAB proteins included in Figure 7B confirms that all of them belong exclusively to members of the order *Acidobacteriales*. It is likely that acidobacteria of this order obtained the fused version of *bcsAB* gene from a common ancestor that received it by lateral transfer from some proteobacteria. The conservative structure of the acidobacterial *bcs* locus clearly indicates that it was wholly transferred from there. The exact origin of the *bcsAB* gene remains unclear: most likely, the fusion of the *bcsA* and *bcsB* genes in proteobacteria has happened several times independently. Even within the same species, several cases have been found of both the presence of individual *bcsA* and *bcsB* genes, and the fused *bcsAB* gene: *Enterobacter cloacae* (GenPept, CZX88434.1 and CZW39489.1), *Escherichia coli* (CQR82949.1 and ASO85436.1), and *Salmonella enterica* (EBH9976197.1 and SUG18009.1).

In the lineage leading to *Acidisarcina polymorpha* SBC82^T^, a non-orthologous displacement of the glycoside hydrolase gene has occurred. As a result, its genome encodes the ACPOL_3660 protein belonging to the GH5_1 subfamily. It should be noted that *Clostridioides difficile* (GenPept, AJP12293.1) and *Clostridium chromiireducens* (OPJ65959.1) have GH5_25 encoding genes in their cellulose synthesis loci (53% amino acid sequence identity). Therefore, we can suppose that GH8/GH5 displacement in the bacterial cellulose synthesis system can provide some adaptive advantage to the cells.

Several acidobacterial species have two paralogs of the fused *bcsAB* gene. Only one of them is located in the cellulose synthesis locus. The second paralog exists in a variable genome context and corresponds to a distinct cluster on the phylogenetic tree indicating its independent evolutionary history. This group of paralogs has appeared in acidobacteria either by ancient gene duplication or by a lateral gene transfer.

In summary, we confirmed the occurrence of genome-encoded capability for cellulose biosynthesis in *Acidisarcina polymorpha* SBC82^T^ and obtained evidence for cellulose presence in the fibrillar envelope produced by this soil acidobacterium. Apparently, these cellulose-made extracellular structures produced by strain SBC82^T^ have a protective function and ensure the survival of this acidobacterium in habitats with harsh environmental conditions. A more detailed analysis of the material composing the cellular envelope that, most likely, includes not only cellulose but also some other polysaccharides, was not feasible due to the extremely slow and poor growth of strain SBC82^T^, which made it impossible to collect sufficient cell material for the conventional chemical analysis. The latter could potentially be applied for the analysis of extracellular material produced by fast-growing acidobacteria of the genera *Terriglobus* and *Granulicella*, which possess genome-encoded capability for cellulose biosynthesis.

## Figures and Tables

**Figure 1 microorganisms-10-02253-f001:**
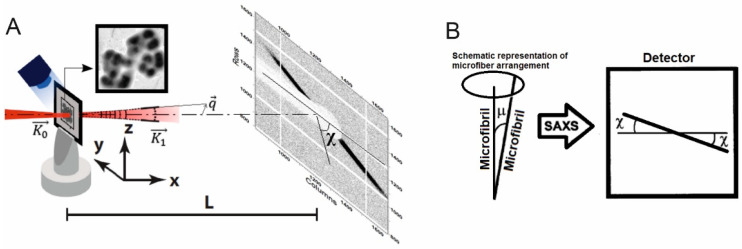
(**A**) Scheme of the SAXS experiment: $\vec{\;K_{0}}$ is a wave vector of the incident beam, $\vec{K_{1}}—$ a wave vector of scattered radiation, $\vec{q}=\vec{K_{1}}-\vec{K_{0}}$ —scattering vector. (**B**) The angle of an individual fibril respectively to the cell axis.

**Figure 2 microorganisms-10-02253-f002:**
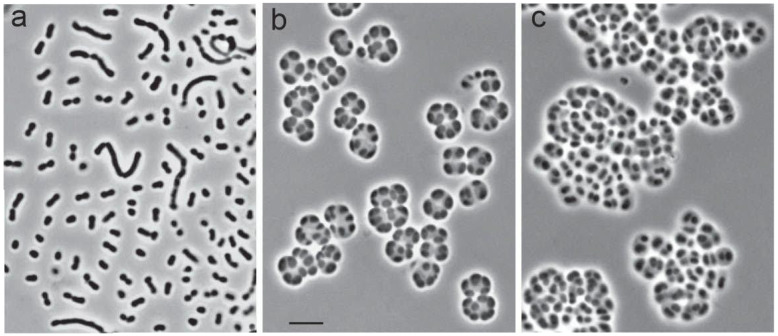
Phase-contrast images of polymorphic cells of *Acidisarcina polymorpha* SBC82^T^ grown for 10 days (**a**), 20 days (**b**), and 40 days (**c**) on the solid medium MA. Bar, 5 μm.

**Figure 3 microorganisms-10-02253-f003:**
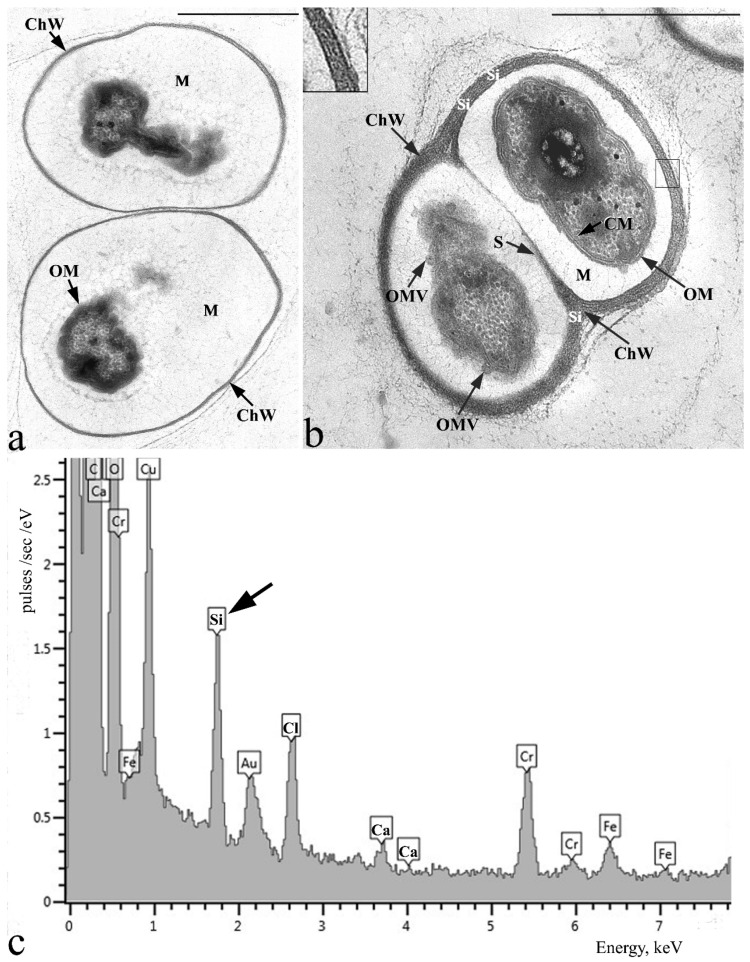
Electron micrograph of the ultrathin sections of 2-week-old (**a**) and 30-day-old (**b**) cells of *Acidisarcina polymorpha* SBC82^T^ grown in the liquid medium MA and X-ray spectrum of chamber wall from 30-days old cell aggregate (**c**). The insert in the upper left corner of (**b**) shows an enlarged fragment of the chamber wall indicated by the box in this figure. ChW—chamber wall; OM—outer membrane; CM—cytoplasmic membrane; S—septum; M—chamber matrix; OMV—outer membrane vesicles; Si—silicon (pointed by arrow in (**c**). Bars (**a**,**b**), 1 μm.

**Figure 4 microorganisms-10-02253-f004:**
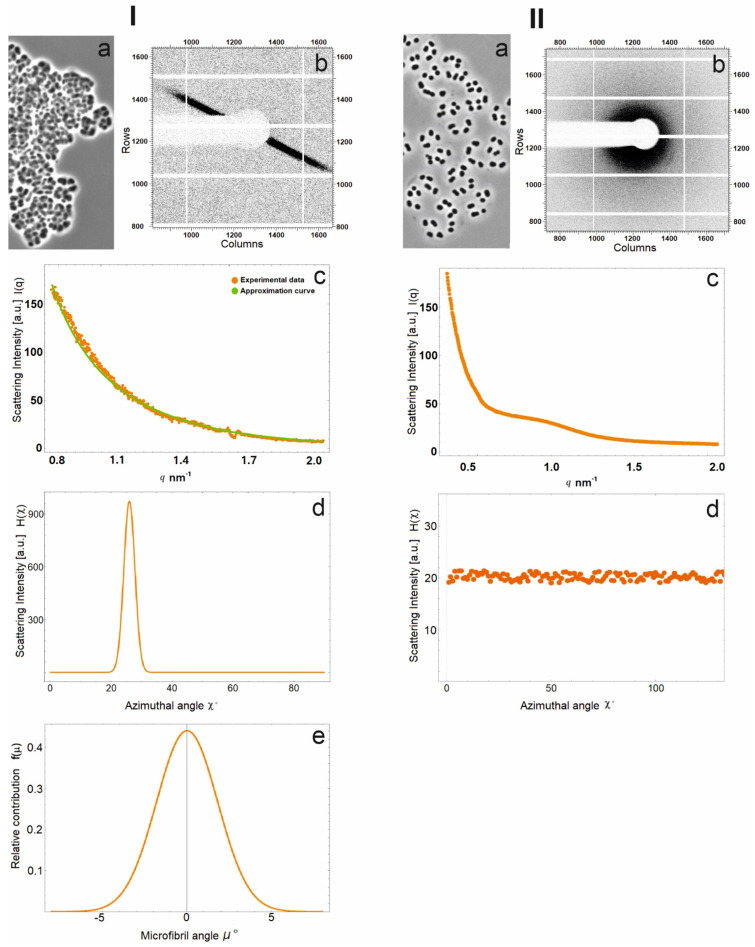
Scattering intensity from 45-day-old (**I**) and 14-day-old (**II**) cultures of *Acidisarcina polymorpha* SBC82^T^. (**a**) Phase-contrast images of bacterial samples. (**b**) 2-D scattering intensity of the samples. (**c**) Scattering intensity *I*(*q_r_*) averaged along azimuthal angle *χ*. (**d**) Scattering intensity *H*(*χ*) averaged along the radial component of scattered vector *q*. (**e**) Calculated probability distribution of individual fibril angel *μ*, zero-based at the cell axis.

**Figure 5 microorganisms-10-02253-f005:**
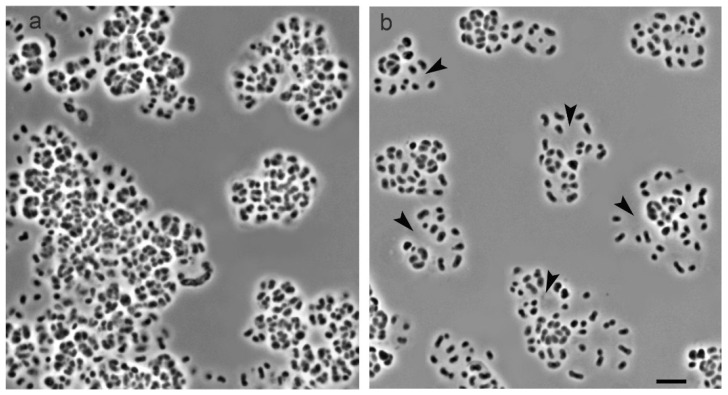
Phase-contrast images of cells aggregates of *Acidisarcina polymorpha* SBC82^T^ after cellulase treatment (**b**) compared to intact aggregates (**a**). Clear zones that appeared due to the degradation of cell conglomerates are indicated by black arrows. Bar, 5 μm.

**Figure 6 microorganisms-10-02253-f006:**
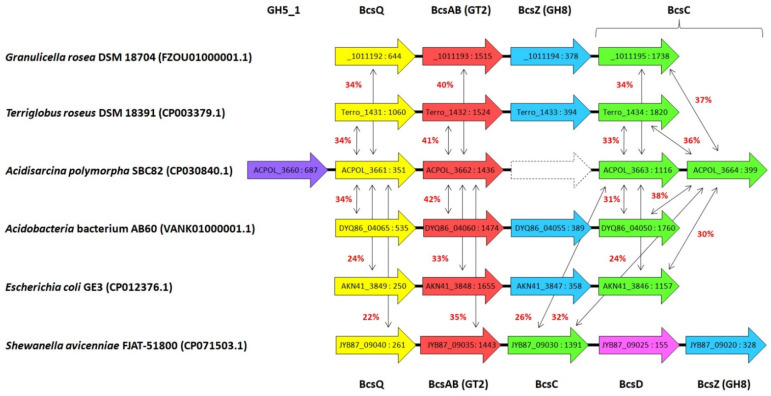
Structure of *Acidisarcina polymorpha* SBC82^T^ cellulose synthase locus and examples of similar loci in other organisms. Strains and the corresponding GenBank accession numbers are shown in the left column. Colored arrows correspond to ORFs, their locus tags and lengths are indicated inside. In the case of *Granulicella rosea* DSM 18704 only the second parts of the locus tags are indicated (SAMN05421770_1011192–SAMN05421770_1011195). The orthologous proteins are indicated by the same color. Names of encoded proteins are indicated in the first and last lines. Red numbers near double-sided thin arrows show the pairwise sequence identity of the *A. polymorpha* proteins and their orthologs. The dotted line is used to indicate the deleted *bcsZ* gene in the *A. polymorpha* SBC82^T^ genome.

**Figure 7 microorganisms-10-02253-f007:**
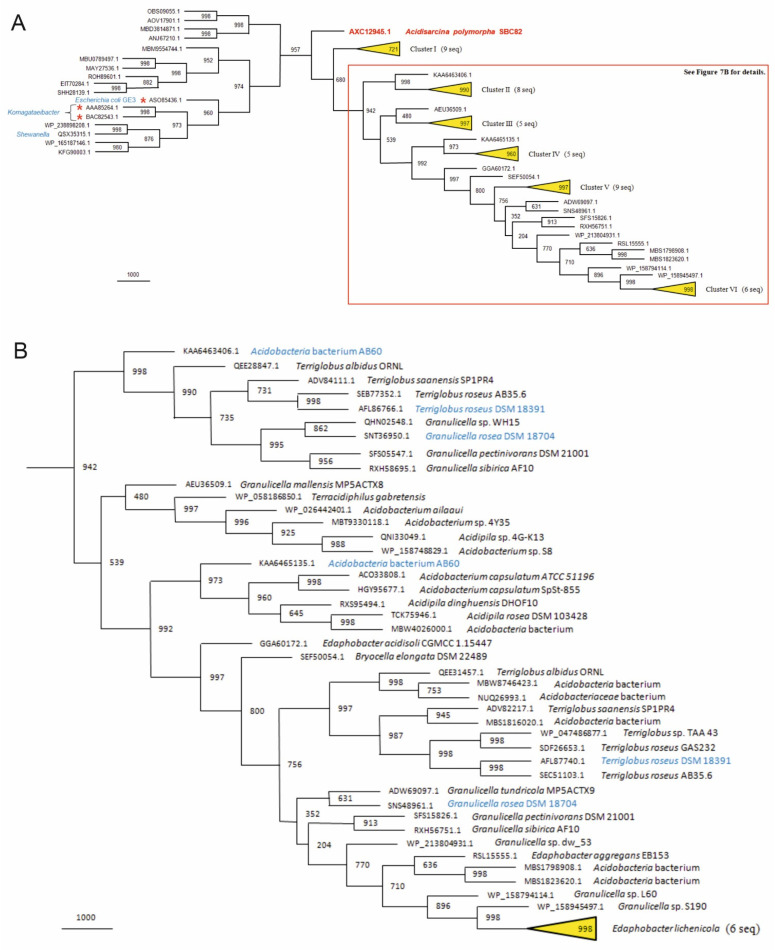
The maximum parsimony phylogenetic tree of the acidobacterial catalytic subunit BcsAB of cellulose synthase. Statistical significance of the tree was assessed by bootstrap analysis; the number of supporting pseudoreplicates (out of 1000) is indicated next to each branching site. (**A**) A simplified scheme of the phylogenetic tree with six undisclosed branch clusters (I–VI) of closely related proteins. The bootstrap support of each cluster is given within the corresponding triangle, and the number of proteins is indicated next to the triangle. Cluster I contains nine proteins encoded by metagenomes (GenPept: MBE7544888.1, MBI1352977.1, MBI4909624.1, MBL0156305.1, MBL8212404.1, MBN8729825.1, MBV8549744.1, PYV80644.1, and PYX09599.1). Cluster VI contains six proteins encoded by different strains of *Edaphobacter lichenicola* (MBB5315980.1, MBB5331838.1, MBB5338266.1, MBB5346008.1, NYF52494.1, and NYF89476.1). Structure of the four clusters (II–V) is shown in (**B**). Protein ACPOL_3662 (AXC12945.1) from *Acidisarcina polymorpha* SBC82^T^ is shown in red. Proteobacterial proteins with experimentally characterized enzymatic activity are indicated by red asterisks. Red rectangle corresponds to the tree fragment shown in (**B**). (**B**) Fragment of the phylogenetic tree in more detail. Species names are indicated for all proteins. Organisms represented in Figure 6 are shown in blue (for both paralogs).

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
