# Peer review of "Building a Cell House from Cellulose: The Case of the Soil Acidobacterium Acidisarcina polymorpha SBC82T"

_microorganisms, 2022, doi:10.3390/microorganisms10112253_

Round 1

Author Response

Comment: L19: The authors did not compare the SAXS of their bacterial cellulose with ‘mature wood samples.’ This conclusion is derived from comparison to values in the literature. I do not think it is acceptable to report this as a conclusion in the abstract.

Response: This sentence has been corrected in order to clarify that the diffraction pattern obtained in our study was similar to those reported earlier for wood samples. Comparing data obtained in different studies is a common practice in scientific research given that the experiments/measurements are performed in a similar way using the same techniques & instrumental basis. The methodology and the parameters of SAXS measurements performed in our study and that of Lichtenegger et al. were the same. Processing of the reciprocal space data obtained in these studies was carried out in the same way. We see no reason why the results of these studies cannot be compared. But we agree that we should have made this clear in the abstract.

 Comment: L36: The most recent standardization of bacterial taxonomy has the Acidobacteriota containing 13 classes (online: https://gtdb.ecogenomic.org/tree?r=p__Acidobacteriota or scholarship: https://academic.oup.com/nar/article/50/D1/D785/6370255). I do not think it is necessary for the authors to follow this phylogeny, since there are Acidobacteriota specific papers that have been cited, but the more current their work is, the better.

Response: The referee did not consider that not all phylogenetic subgroups of acidobacteria may have been represented by available genome sequences. The pool of acidobacterial 16S rRNA gene sequences by far exceeds the pool of genome sequences.

 Comment: L60: Capitalize ‘acidobacteria’?

Response: Corrected.

Comment: L86: It would be best to cite the primary literature here, since specific examples are given.

Response: Done. We have added 4 additional references.

 Comment: L93 - 98: This information is tangential to the topic of your paper. It is too much detail and should be removed. You might introduce the function of GH8 and GH5 here, but it is probably best to add that information to the Discussion, since you discovered the unique bscZ configuration based on your observations (not based on an existing expectation prior to your study).

Response: We have reduced this text fragment. However, presenting this short info is of importance for the reader since family 2 of glycosyltransferases (GT2) and the GH5_25 subfamily endoglucanase are further addressed in this text section.

 Comment: L179: Please provide a quick explanation for how the conclusion about the pseudo genes was reached.

Response: The proteins of Candidatus Sulfotelmatomonas gaucii (GenPept, SPE19250.1) and Granulicella aggregans M8UP14 (MBB5059869.1) were excluded from further analysis because they are incomplete. The protein of Candidatus Sulfotelmatomonas gaucii (GenPept, SPE19250.1) does not have the N-terminal part and the protein of Granulicella aggregans M8UP14 (MBB5059869.1) has a deletion in the middle part.

Comment: L203: “On ageing…” <- awkward, try: “Over time…”

Response: Corrected as recommended.

Comment: L229: “Possibly, these vesicles…” <- Speculation like this can be presented in the Discussion section, but not the Results section. Please move.

Response: This sentence has been omitted.

Comment: L234: “…thus suggesting protective function…” <- Again, this speculation is welcome in the Discussion, but not presented as a result.

Response: This wording has been omitted from the Results.

Comment: L306: How can the authors be sure the matrix is entirely comprised of cellulose? It would have been more compelling result to see all the cells freed of their sacculi and dispersed in the medium. I recognized that a 2 hr incubation with an endoglucanase will probably not degrade all the fibers, so the image obtained suggests that the material is mostly cellulose. Still, it is not incontrovertible evidence that cellulose is the primary or sole component of the extracellular matrix.

Response: We have carefully re-checked our manuscript. To be fair, there was no statement that “the matrix is entirely comprised of cellulose” in the text. We only stated “the presence of cellulose in the extracellular saccular structures of strain SBC82T”. We absolutely agree that cellulose is not the only component of these extracellular structures. For sure, they include some other polysaccharides as well. We have added this statement in the Discussion.

Comment: L329: How confident are the authors that a protein with < 40% identity is homologous? This seems low.

Response: Protein homology (a common evolutionary origin) is always inferred from statistical analysis (E-value) and not from percent of sequence identity.

Comment: L344: Capitalize ‘proteobacteria’. L379: Capitalize ‘acidobacteria’

Response: In these sentences, we address not the taxa but members of these taxonomic groups.

 Comment: L448: A cellulase in Rhizobium was shown to modify bacterial cellulose in ways that modulated surface colonization of plant roots. Does your GH5 match to this gene (‘Cel2’): doi:10.1186/1475-2859-11-125

Response: This Rhizobium leguminosarum cellulase (GenPept, CAD90973.1) belongs to the GH8 family but not to the GH5 family. Please see the list at http://www.cazy.org/GH8_characterized.html.

Reviewer 2 Report

The manuscript of Belova and coworkers is a valuable report on the structure of the cell-house of the recently described bacterium Acidisarcina polymorpha. Unfortunately, the half of the manuscript is merely based on literature survey, and hypotheses, not on experiments. The reviewer is aware of the fact that the cultivation of this bacterium might be a nightmare, though, ecological adaptation questions ought to be based on experiments.

However, let us see the facts in order of the appearance.

- The title is misleading, since the ecological adaptation is not based on experimentation, simply on literature review, and confabulation. Why there were e.g. not cultivation experiments made to test the hypotheses. E.g., simply cultivate at low pH, and check whether formation of the chamber starts earlier than at neutral pH. Such simple experiments could at least be tried.

- The abstract is concise, though here again, the last sentence is based on speculation and not facts.

- In the Materials and Methods part the cultivation times have to be added in the subchapters 2.1. and 2.2.

- lines 172-173. Please add the correct concentration of the cellulase solution.

- line 203. I do not really feel based on the photo, that cells are regular!

- line 208. Please add the abbreviation of the culture medium.

- line 217. The name of the medium fails.

- line 218. Please add the abbreviation for Si - since on cannot easily understand why the abbreviation Si is present in the chamber cross section.

- line 223. Where you know this? Unfortunately, this cannot be seen. Please add a higher resolution image.

Figure 5. Please add arrows into picture b, where this „clearance” seems to be evident.

- lines 440 - 447. What about Thermotogae?

- lines 460 - 465. Why was the origin of bcsZ not examined in detail?

Author Response

Comment: The title is misleading, since the ecological adaptation is not based on experimentation, simply on literature review, and confabulation. Why there were e.g. not cultivation experiments made to test the hypotheses. E.g., simply cultivate at low pH, and check whether formation of the chamber starts earlier than at neutral pH. Such simple experiments could at least be tried.

Response: We have modified the title to avoid overstatements and omitted the wording related to ‘ecological adaptation’. Our original design of this story was exactly the same as suggested by the referee. We have started with cultivation at different pH and temperature values. Unfortunately, no pronounced difference with regard to the cell chamber formation was noticed within the range of pH (4.5-6.5) and temperatures (10-25C) suitable for growth of these bacteria. We also tried more extreme conditions (pH 3.8-3.9 and 5C) but failed to receive any growth. After one year of fruitless efforts, we changed our research strategy and were able to assemble the currently presented story. We assure the referee that any cultivation-based studies of these bacteria are extremely difficult and time consuming.

Comment: The abstract is concise, though here again, the last sentence is based on speculation and not facts.

Response: We have replaced “apparently’ with ‘presumably’ to indicate suggestive character of this statement.

Comment: In the Materials and Methods part the cultivation times have to be added in the subchapters 2.1. and 2.2.

Response: Done.

Comment: lines 172-173. Please add the correct concentration of the cellulase solution.

Response: Done.

Comment: line 203. I do not really feel based on the photo, that cells are regular!

Response: The definition ‘regular’ has been omitted.

Comment: line 208. Please add the abbreviation of the culture medium.

Response: Done.

Comment: line 217. The name of the medium fails.

Response: This has been corrected.

Comment: line 218. Please add the abbreviation for Si - since on cannot easily understand why the abbreviation Si is present in the chamber cross section.

Response: Done.

Comment: line 223. Where you know this? Unfortunately, this cannot be seen. Please add a higher resolution image.

Response: Ok, we have included an insert in Figure 3b, which shows enlarged view of a specific part of the chamber wall.

Comment: Figure 5. Please add arrows into picture b, where this „clearance” seems to be evident.

Response: Done.

Comment: lines 440 - 447. What about Thermotogae?

Response: The sheath-like envelopes of Thermotoga-related bacteria are completely different from the cell chamber of Acidisarcina. Thermotoga’s sheath (or ‘a toga’) is an outer cell membrane, so that each cell is surrounded by a sheath. The chamber of Acidisarcina is clearly separated from the cell and has nothing to do with outer cell membrane. The chamber can accommodate several cells, so that cells divide inside this structure.

Comment: lines 460 - 465. Why was the origin of bcsZ not examined in detail?

Response: It was not examined in detail since there is no bcsZ gene in the genome of Acidisarcina polymorpha SBC82.